# Bayesian data assimilation for estimating instantaneous reproduction numbers during epidemics: Applications to COVID-19

Xian Yang[1,2☯], Shuo Wang[2,3,4☯], Yuting Xing[2☯], Ling Li[5], Richard Yi Da Xu[6], Karl J. Friston[7], Yike Guo[1,2]*

1 Department of Computer Science, Hong Kong Baptist University, Hong Kong Special Administrative Region, China, 2 Data Science Institute, Imperial College London, London, United Kingdom, 3 Digital Medical Research Center, School of Basic Medical Sciences, Fudan University, Shanghai, China, 4 Shanghai Key Laboratory of Medical Image Computing and Computer Assisted Intervention, Shanghai, China, 5 School of Computing, University of Kent, Kent, United Kingdom, 6 Faculty of Engineering and Information Technology, University of Technology Sydney, Sydney, Australia, 7 Institute of Neurology, University College London, London, United Kingdom

☯ These authors contributed equally to this work.
* yikeguo@hkbu.edu.hk

**Data Availability Statement:** We are releasing DARt as open-source software for epidemic research and intervention policy design and monitoring. The source code of our method and

## Abstract

Estimating the changes of epidemiological parameters, such as instantaneous reproduction number, $R_t$, is important for understanding the transmission dynamics of infectious diseases. Current estimates of time-varying epidemiological parameters often face problems such as lagging observations, averaging inference, and improper quantification of uncertainties. To address these problems, we propose a Bayesian data assimilation framework for time-varying parameter estimation. Specifically, this framework is applied to estimate the instantaneous reproduction number $R_t$ during emerging epidemics, resulting in the state-of-the-art 'DARt' system. With DARt, time misalignment caused by lagging observations is tackled by incorporating observation delays into the joint inference of infections and $R_t$; the drawback of averaging is overcome by instantaneously updating upon new observations and developing a model selection mechanism that captures abrupt changes; the uncertainty is quantified and reduced by employing Bayesian smoothing. We validate the performance of DARt and demonstrate its power in describing the transmission dynamics of COVID-19. The proposed approach provides a promising solution for making accurate and timely estimation for transmission dynamics based on reported data.

## Author summary

Monitoring the evolution of transmission dynamics is of great importance in response to the COVID-19 pandemic. The transmission dynamics of infectious disease is described by epidemiological models, but the model parameters may vary substantially due to differences in government intervention policies. Existing methods on estimating time-varying epidemiological parameters face problems such as lagging observation, averaging

our web service are publicly available online
(https://github.com/Kerr93/DARt).

**Funding:** The author(s) received no specific
funding for this work.

**Competing interests:** The authors have declared
that no competing interests exist.

inference, and unreliable uncertainty. To address these issues, we have proposed the
Bayesian data framework to provide a timely estimate with credibility interval. We have
developed the 'DARt' system to monitor the instantaneous reproduction number $R_t$ from
daily COVID-19 reports. The accuracy and robustness of our system are validated in
numerical simulations and in retrospective analyses of real-world scenarios. Our system
provides the insights of impacts of different intervention polices and highlights the effec-
tiveness of undergoing mass vaccination.

## Introduction

Epidemic modelling is important for understanding the transmission dynamics and respond-
ing to the emerging COVID-19 pandemic [1–8]. Since the pilot work by Kermack and McKen-
drick [9], various epidemic models with different governing equations have been developed to
describe the transmission dynamics of infectious diseases [10]. For common infection diseases
such as influenza, the epidemiological parameters are related to the nature of the virus and
treated as constants during the epidemic outbreak. These models are not applicable to the
emerging COVID-19 pandemic where extensive government control measures have been
implemented and continuously revised. Due to the impacts of control measures, the epidemio-
logical parameters (e.g., infection rates) linked to human behaviours could change substan-
tially. In particular, the instantaneous reproduction number $R_t$, defined as the expected
number of secondary cases occurring at time $t$, divided by the number of infected cases scaled
by their relative infectiousness, has drawn extensive attention [11]. Estimating such time-vary-
ing parameters from epidemiological observations (e.g., daily report of confirmed cases) is use-
ful for nowcasting transmission [12], for retrospectively assessing intervention impacts and for
developing vaccine strategies [6,13,14]. All the applications depend on a reliable system for
estimating the time-varying parameters with accuracy and timeliness. Imprecise estimation or
inappropriate interpretation could feed misinformation. Several systems [10,15–18] have been
proposed to estimate the time-varying epidemiological parameters in practice; however, this
remains a challenging task due to the following issues [19]:

a. **Lagging observations.** Given a mathematical model of transmission dynamics, to infer the
time-varying parameters, the number of infections should be the ideal observable data.
However, the actual infection number is often unknown and can only be inferred from
other epidemiological observations (e.g., the daily confirmed cases). Such observations are
lagging behind the infection events due to inevitable time delays between the infections of
individual patients and the detection of the cases (e.g., days for symptom onset [20]). Direct
parameter estimation based on lagging observations without adjusting for the time delay
results in the temporal inaccuracy of estimates [19,21]. To address this problem, a two-step
strategy, first estimating infections from epidemiological observations with a temporal
transformation followed by parameter estimation, has been commonly used in practice
[21]. The simple temporal shift of observations by the mean observation delay turns out
insufficient for the relatively long observation delay or the rapidly changing transmission
dynamics, which are seen in the COVID-19 pandemic [21]. Backward convolution method
(i.e., subtracting time delay, with a given distribution, from each observation time) leads to
an over-smooth reconstruction of the infection number and bias for parameter estimation
[10]. Deconvolution methods [22] through inversing the observation process are mathe-
matically more accurate but sensitive to the optimisation procedures (e.g., stopping crite-
rion) of the ill-posed inverse problem. In addition, the estimated result of infection number

is often calculated as a point estimate, thereby overlooking the uncertainty from the observation process is neglected [23]. Taking an approach alternative to the two-step strategy, we are investigating a new Bayesian method that could jointly estimate both infection number and epidemiological parameters with uncertainty by explicitly parameterising the observation delay.

b. **Averaging inference.** There are two general paradigms to deal with the challenge of estimating time-varying parameters: 1) reformulating the problem into an inference of static or quasi-static parameters, so that various methods for static parameter estimation can be used; 2) developing inference methods for explicit time-varying parameter estimation. For the first approach, the time-varying parameter is usually parameterised with several static parameters (e.g., the initial value and the exponential decay rate [17]). When adopting the quasi-static method, it is assumed that such a slow evolution of the parameter that could be treated as static within a short period. For example, Cori et al. [16] proposed a sliding-window method 'EpiEstim' using a segment of observations for the averaging inference of $R_t$, assuming $R_t$ remains the same within the sliding window. But this assumption does not apply to the rapidly changing transmission dynamics, in which the window size affects the accuracy. Best practices of selecting the sliding window are still under investigation [21]. Instead of adopting a local sliding window, Flaxman et al. [13] defined several periods according to the dates of intervention measures, assuming a constant $R_t$ within each period. This approach requires additional information about the intervention timeline, which could be inaccurate, and does not capture the abrupt change of $R_t$. In contrast to these window-based methods, data assimilation [24] is a window-free alternative approach that has been less explored for parameter estimation in computational epidemiology. Applying sequential Bayesian inference [25,26], data assimilation supports instantaneous updating of model states upon the availability of new observation data. The Bayesian model selection mechanism [27] can also be used for modelling the switching transmission dynamics under interventions, thereby avoiding the drawback of averaging inference. Different from the common compartment models [28–30] used in concurrent data assimilation studies of COVID-19 modelling, we use the renewal process taking into account for the changing infectiousness of the virus during the infection period. Moreover, we propose the Bayesian smoothing scheme that allows the correction of historical estimates based on subsequent observations.

c. **Quantification of uncertainty.** The credibility of parameter estimation is as important as the estimate itself, especially for policymaking. The uncertainty comes from multiple sources, including the intrinsic uncertainties of epidemic modelling, data observation and inference processes. Firstly, the uncertainty of epidemiological models affects the final estimates. For example, $R_t$ estimation is found to be sensitive to the assumed distribution of generation time intervals [21]. Secondly, the uncertainty, resulting from systematic errors (e.g., weekend misreporting) and random errors (e.g., spike noise) in the observation processes should be properly quantified. During the COVID-19 pandemic, for example, we have seen different reporting standards and time delay across countries and regions, with different levels of uncertainty. Thirdly, the uncertainty could be enlarged or smoothed in the inference processes. For example, the use of a sliding window could smooth the parameter estimation but may simultaneously miscalculate the uncertainty, due to the overfitting within the sliding-window. To provide reliable credibility intervals (CrI) of parameter estimates, the three aforementioned types of uncertainty should all be considered and reported as part of the final estimates. A state-of-the-art package for $R_t$ estimation, EpiEstim (Version 2) [31], allows users to account for the uncertainty from epidemiological parameters by

resampling over a range of plausible values. However, the uncertainty from imperfect observations and the side effects associated with the sliding window cannot be processed by this tool. Recently, 'EpiNow' [18] was proposed to integrate the uncertainty of observation process, but the inference is still based on the sliding window. In this work, we deal with model and data uncertainty in the data assimilation framework [24] with a Bayesian smoothing mechanism to enable both the latest and historical observations to continuously integrate into inference flow, thereby alleviating spurious variability of estimations.

In order to tackle these practical issues, we propose a comprehensive Bayesian data assimilation system, for estimating time-varying epidemiological parameters together with their uncertainty. In particular, we focus on the joint estimation of infection numbers and $R_t$ as a real-world application. Compared to the Bayesian approach for estimating the basic reproduction number $R_0$ at the beginning stage of an epidemic break [32], the sequential updating scheme is developed in our system. The evolution of the transmission dynamics is described by a hierarchical transition process, which is informed by newly data formulated with explicit observation delay. A model selection mechanism is built in the transition process to detect abrupt changes under interventions.

## Results

### 1. Bayesian data assimilation for epidemiological parameter estimation

We propose a Bayesian data assimilation approach, as illustrated in Fig 1, to estimate the time-varying parameters based on epidemiological observations. This framework is applicable to various epidemic models when the governing equations and observation functions are available. Given an epidemic model (e.g., renewal process), we can construct a latent state $X_t$ at time $t$ which consists of the time-varying variables and parameters of the governing equations. The epidemiological observations $C_{1:T}$ up to the latest observation time $T$ are made during the observation process of the latent state $X_t$. The problem of estimating the time-varying parameters can be formulated as a Bayesian inference problem of $p(X_t|C_{1:T})$ for each time step $t$. In contrast to inferring the 'pseudo' dynamics (i.e., reformulating into a static/quasi-static problem), our method directly estimates the 'real' dynamics by assimilating information from the observations for the epidemic model forecast.

As illustrated in Fig 1, the Bayesian data assimilation has two phases: forward filtering and backward smoothing. The forward filtering uses the up-to-date prior from the state transition model and the likelihood determined by the latest observation to update the current latent state, by computing its posterior distribution following Bayes' rule. For the implementation of this Bayesian updating process, we adopt a particle filter method [26] to efficiently approximate the posterior distribution. The backward smoothing works by looking back to refine the previous state estimation when more observations are accumulated to reduce the uncertainty of parameter estimation. That is, the estimation of latent state at a time $t$ is smoothed retrospectively, given all observations available till time $T$ ($T>t$). Please refer to the Method section for more detailed explanations.

### 2. DARt: A data assimilation system for $R_t$ estimation

To apply the proposed Bayesian data assimilation approach in a real-world problem, we developed the 'DARt' (Data Assimilation for $R_t$ estimation) system for the $R_t$ estimation. The transmission dynamics is described by the governing equations of the renewal process, where $R_t$ is the key epidemiological parameter driving the number of incident infections $j_t$. We construct the latent state $X_t$ including the variable $j_t$, the time-varying parameter $R_t$ and the auxiliary

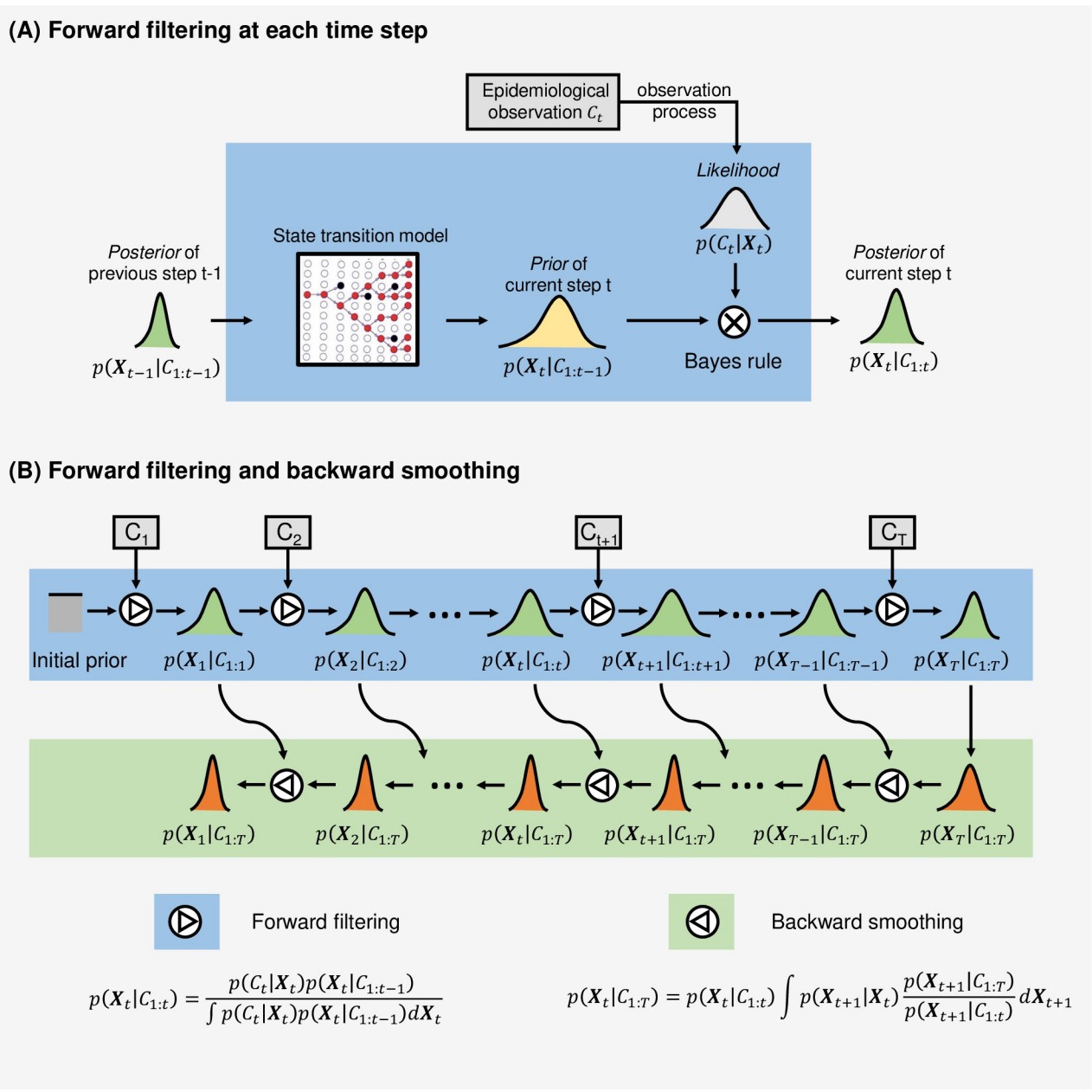

**Fig 1.** Illustration of the inference of Bayesian data assimilation system for time-varying parameter estimation. The latent state $X_t$ includes the variables and parameters of an epidemic model to be estimated. The epidemiological observation is denoted as $C_t$, and is linked to the latent state via the observation function. For each time step, the estimation of the latent state $p(X_t|C_{1:t})$ is constantly updated according to ongoing reported observations using sequential Bayesian updating with forward filtering and backward smoothing. **(A) Forward filtering at each time step**. The posterior state estimation $p(X_{t-1}|C_{1:t-1})$ estimated from previous step $t-1$ is transformed as the *prior* $p(X_t|C_{1:t-1})$ for the current step $t$, calculated from the state transition model as detailed in the Method section. Together with the *likelihood* $p(C_t|X_t)$ obtained from epidemiological observation at the current step, the *posterior* of the current step $p(X_t|C_{1:t})$ is estimated. At the same time, as shown in **(B)**, **backward smoothing** is used to compute $\{p(X_t|C_{1:T})\}_{t=1}^{T}$, taking account of all the observations $C_{1:T}$ up to the time $T$ by applying a Bayesian smoothing method (see the Methods section for more informaiton).

variable $M_t$ for switching dynamics. Notably, $M_t$ is to indicate the switching dynamics of the epidemiological parameter: $M_t = 0$ indicates smooth changes, while $M_t = 1$ indicates an abrupt change. As detailed in the Methods section, the dynamics of the latent state $X_t$ can be described

**Fig 2. Validation experiment of the DARt system on simulated data.** First, the ground-truth $R_t$ sequence is synthetic using piecewise Gaussian random walk split by several abrupt change points. The sequence of incident infection $j_t$ is simulated based on a renewal process parameterised by the synthetic $R_t$. The observation process includes applying a convolution kernel that represents the probabilistic observation delay to obtain the expected observation $\bar{C}_t$ and adding Gaussian noise that represents the reporting error to obtain the noisy 'real' observation $C_t$. The inputs (in grey) to the DARt system are the distributions of generation time, observation kernel and simulated noisy observation $C_t$. The system outputs are the estimated $\hat{R}_t$, estimated $\hat{j}_t$ and change indicator $\hat{M}_t$. These outputs are compared with the synthetic $R_t$, $j_t$ and the time of abrupt changes. Also, the observation function is applied to the estimated $\hat{j}_t$ to compute the estimated observation $\hat{C}_t$ with uncertainty, which is compared to the 'real' observation.

using a hierarchical transition model, where $R_t$, $j_t$ and $M_t$ can be estimated by DARt. Under the modelling of convolutional observation process, we test the capacity of DARt with different observation inputs and kernels. The performance of the DARt system is validated and compared to that of the state-of-the-art EpiEstim and EpiNow2 systems through simulations and real-world applications. The results confirm its power of estimation and adequacy for practical use. We have made the system available online for broad use in $R_t$ estimation for both research and policy assessment.

**Validation through simulation.** Due to the lack of ground-truth $R_t$ in real-world epidemics, we conduct a set of simulation experiments by using synthetic data for validation. Fig 2 illustrates the design of simulation experiments where a synthetic $R_t$ is adopted as the ground truth to validate its estimated $\hat{R}_t$. We also estimated $R_t$ using the state-of-the-art $R_t$ estimation package EpiEstim [31] and EpiNow2 [18] to compare the effectiveness in overcoming the three aforementioned issues (i.e., lagging, averaging and uncertainty).

**Experimental settings.** In the simulation experiments, we compare the performance of DARt with that of two comparative methods: EpiEstim and EpiNow2. These two models are applied under their default settings with a 7-day smoothing window. When applying EpiEstim, we adopt the two-step strategy that shifts $C_t$ backwards in time by the median observation delay (5 days in the simulation). As the current implementation of EpiNow2 (https://github.com/epiforecasts/EpiNow2) only supports Gamma and Lognormal distribution as time delays, we set the generation time distribution to be a Gamma distribution with shape and scale equal to 4.44 and 1.89,respectively (obtained by fitting the Weibull distribution reported by Ferretti et al. [3] using the Gamma distribution). With the simulated $R_t$ curve and the generation time distribution, we follow the renewal process to simulate the infected curve $j_t$ (initialized to be 1). Then, the observation curve of onset cases $\bar{C}_t$ is generated using the incubation time

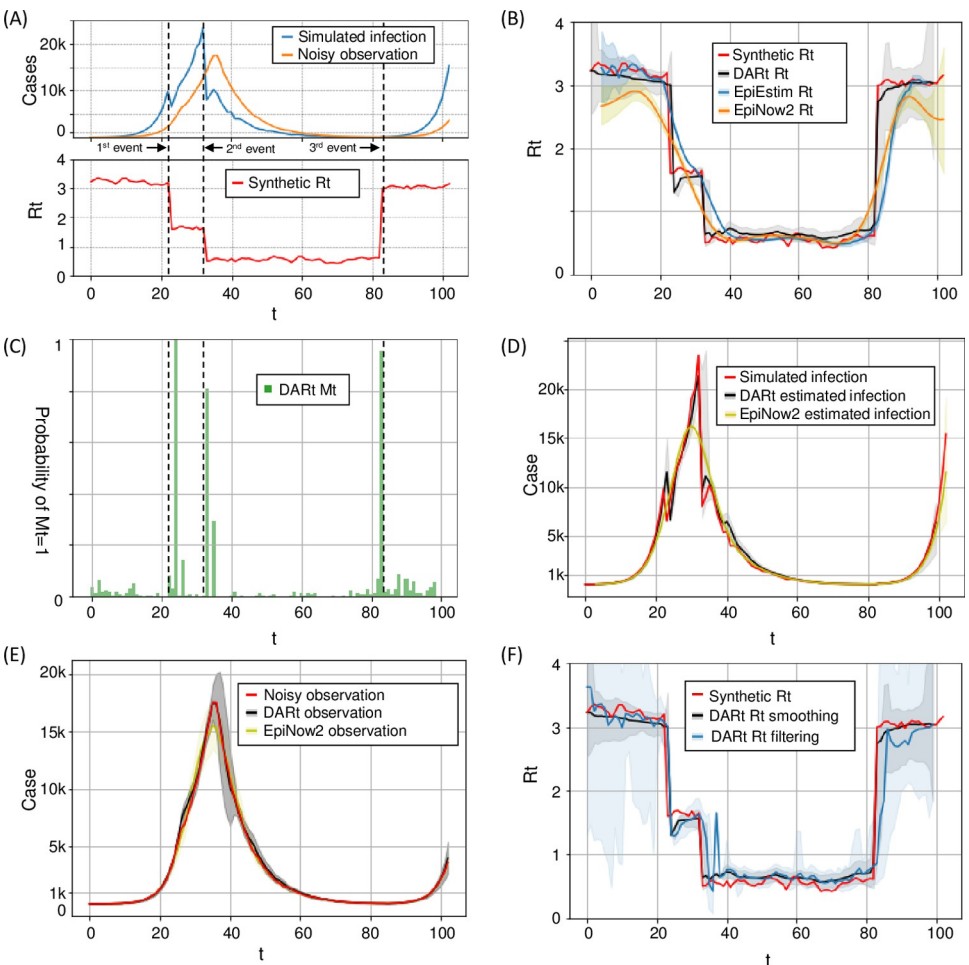

**Fig 3. Simulation results. (A)** Synthetic $R_t$, simulated $j_t$ and $C_t$ curves. **(B)** shows the comparison of the synthetic $R_t$ (in red) with estimated $R_t$ curves from DARt, EpiEstim and EpiNow2. **(C)** shows the estimated $M_t$ from DARt to indicate sharp changes of $R_t$. **(D)** shows the simulated $j_t$, $j_t$ from DART, and $j_t$ from EpiNow2. **(E)** compares the distributions of estimated $C_t$ from DARt and EpiNow2 with the simulated $C_t$ curve with 95% CrI. **(F)** compares the DARt estimated $R_t$ results with and without smoothing.

distribution [3] (i.e., the lognormal distribution with log mean and standard deviation of 1.644 and 0.363 days respectively) as the observation time delay. Similar to the experiments in other related work [12,13], all comparative models start estimation when the daily observation exceeds a threshold number, which is set to be 10 in our experiments.

Fig 3A shows the synthetic $R_t$ curve following a piece-wise Gaussian random walk that mimics the scenario of two successive interventions and one resurgence. To approximate the early stage of exponential growth, the simulation starts with $R_0 = 3.2$ (i.e. the basic reproduction number) and follows a Gaussian random walk $R_{t+1} \sim Gaussian(R_t, (0.05)^2)$. At $t = 23$, we set $R_{23} = 1.6$ indicating the mitigation outcome of soft interventions. After soft interventions, the epidemic is still being uncontrolled with the evolution of $R_t$ resuming to the Gaussian random walk as above. At $t = 33$, $R_t$ decreased abruptly to a value under 1, where we set $R_{33} = 0.5$ to indicate the suppression effects of intensive interventions (e.g., lockdown). After the epidemic is controlled for a while, one outbreak happens at $t = 83$ with $R_{83} = 3$. The evolution of $R_t$ after this resurgence follows the random walk for a few days.

To simulate the real-world noisy observations, we added Gaussian noise with zero mean and standard deviation equal to $N$ times of $\bar{C}_t$. The results presented in the next section are obtained with $N = 1$. To further investigate the performance of all comparable models under different levels of noise, we also show the results when $N$ is chosen from {0,1,2,3} in Fig F in S1 Text. Notably, in the rest of this main manuscript both the generation and incubation time distributions are truncated and normalised, i.e. values smaller than 0.1 are discarded. Sensitivity analysis has been done and reported in Fig G in S1 Text showing impacts of different choice of threshold. Fig H in S1 Text represents the uncertainties resulted from different settings of time distributions.

**Simulation results.** All simulation results are presented in Fig 3 and discussed as follows.

- **Correctness of $R_t$ estimation**: Fig 3B compares the synthetic $R_t$ with the estimated $R_t$ from DARt, EpiEstim and Epinow2. We can see that $R_t$ from DARt matches the synthetic $R_t$ better than that from the other comparative methods with relatively less degree of fluctuations and faster response to abrupt changes. The results demonstrate that the proposed model can mitigate the influence of noisy observations and overcome the weakness of averaging. The probabilities of having abrupt changes are captured by $M_t$ as shown in Fig 3C. Even with observation noise, DARt can still detect abrupt changes.

- **Correctness of $j_t$ estimation**: Fig 3D shows the simulated $j_t$, DARt estimated $j_t$, and Epi-Now2 estimated $j_t$. We can find that the DARt estimated $j_t$ with 95% CrI match well the simulated $j_t$. In contrast, the estimated $j_t$ curve from Epinow2 is over-smoothed such that sharp changes in the simulated $j_t$ cannot be captured by Epinow2. In particular, the peak value of $j_t$ from EpiNow2 deviates greatly from the simulated value.

- **Accuracy in recovering observations $C_t$**: Fig 3E compares the distributions of reconstructed $C_t$ from DARt and that of EpiNow2. We can find that compared with $C_t$ from EpiNow2, $C_t$ from DARt with 95% CrI can generally match well with the simulated $C_t$.

- **Effectiveness of DARt smoothing**: Fig 3F illustrates the effectiveness of backward smoothing by comparing the DARt estimated $R_t$ results with and without smoothing, showing the expected smoothing effect of estimated $R_t$ with reduced CrI. It is clear that the results from DARt without smoothing are affected by local fluctuations, which are probably due to observation noises. With the introduction of smoothing, both the uncertainties and local fluctuations are reduced.

**Applicability to real-world data.** We applied DARt to estimate $R_t$ in four different regions during the emerging pandemic. Each region represents a distinct epidemic dynamic, allowing us to test the effectiveness and robustness of DARt in different scenarios. **1) Wuhan**: When the outbreak of COVID-19 happened in Wuhan, the government responded with very stringent interventions such as a total lockdown. By studying its $R_t$ evolution, we can check the capability of DARt in detecting the abrupt changes of $R_t$. **2) Hong Kong**: The daily increase of reported cases in Hong Kong has been remained at a low level for most of the time with the maximum daily cases under 200. As no stringent interventions have been introduced in such a city with high-density population, Hong Kong offers an ideal scenario for studying the change of $R_t$ under mild interventions. **3) United Kingdom:** The daily infection number in the UK changed significantly this year, varying from 2,000 to 6,000. UK is one of the first countries initiating mass immunisation campaign; therefore, its instantaneous $R_t$ is a useful metric for checking the efficacy of vaccination in real world on the way towards 'herd immunity'. **4) Sweden:** Sweden is a representative of countries that have less stringent intervention policies; it

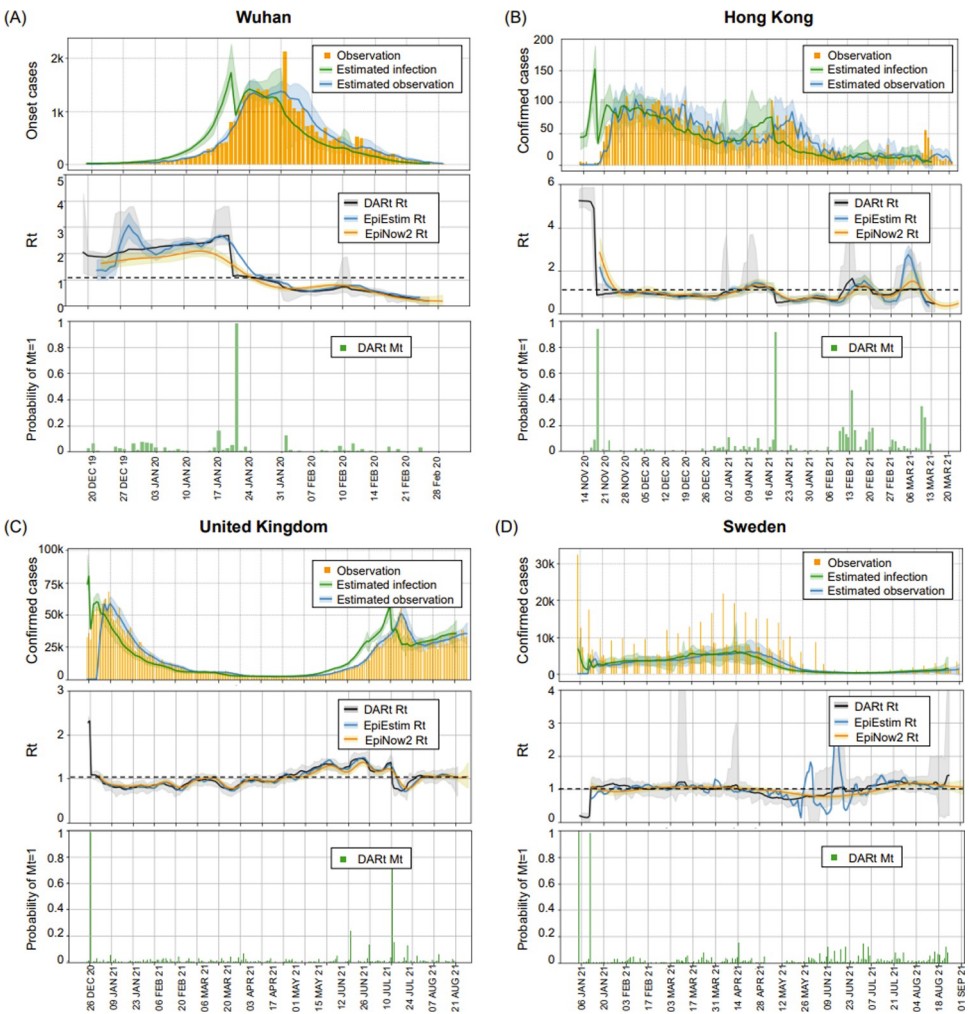

**Fig 4.** Epidemic dynamics in Wuhan **(A)**, Hong Kong **(B)**, United Kingdom **(C)** and Sweden **(D)**. The top row of each subplot shows the number of daily observations (in yellow), the estimated daily observations (in blue) and the estimated daily infections (in green). The middle row compares the DARt $R_t$ estimation (in black) with the EpiEstim results (in blue) and EpiNow2 (in yellow). The distributions of all estimated $R_t$ are with 95% CrI. The bottom row shows the probabilities of having abrupted changes ($M_t = 1$) (in green bars).

has a clear miss-reporting pattern repeated periodically. This makes Sweden an ideal case to examine the robustness of DARt with considerable observation noises.

**Epidemic dynamics in these four regions.** The inference results for $R_t$ and reconstructed observations for these four regions are shown in Fig 4. For Wuhan, the observation data are the number of onset cases compiled retrospectively from epidemic surveys, while for Hong Kong, UK and Sweden, the observation data are the number of reported confirmed cases. Notably, we use the onset-to-confirmed delay distribution from [33] together with the distribution of incubation time proposed in [3] to approximate the observation delay. As the ground-truth $R_t$ is not available, we validate the results by checking whether the estimated distributions of observation match well with the observation curve of $C_t$. As shown in the top panel of each subplot in Fig 4, the CrIs of estimated $C_t$ distributions (in blue) match most parts of the original observations (in yellow), confirming the reliability of our $R_t$ estimation.

Fig 4A shows the results of testing using Wuhan's onset data [1]. We observe that there was a sharp decrease in Wuhan's $R_t$ after January 21, 2020, which is also illustrated by $M_t$ as the probability of abrupt changes peaked at this time (in green bars). A strict lockdown intervention has been enforced in Wuhan since January 23, 2020. This sharp decrease in Wuhan's $R_t$ is likely to be the result of this intervention. The small offset between the exact lockdown date and the time of sharp decrease might be due to noisy onset observations and approximated incubation time distribution. After the lockdown, $R_t$ decreased smoothly, indicating that people's increasing awareness of the disease and the precaution measures taken had made an impact. Since the beginning of February 2020, the value of $R_t$ remained below 1 for most of the time with the enforcement of quarantine policy and increases in hospital beds to accept all diagnosed patients. It is noted that the onset curve has a peak on February 1, 2020, due to a major correction in the reporting standard. Neither $R_t$ nor $j_t$ curve from our model were severely affected by this fluctuation, highlighting the robustness of our model thanks to the smoothing mechanism. The results from Wuhan suggest that our switching mechanism can address the issue of averaging and automatically detect sharp changes in epidemiological dynamics. The results from DARt are also compared with results from EpiEstim and EpiNow2. EpiEstim generates results with significant local variations and delays, while EpiNow2 can derive a smooth $R_t$ curve with no obvious delays. However, the immediate impact of lockdown cannot be well detected by EpiNow2.

Fig 4B shows the inferred results from Hong Kong that reported confirmed cases [34] during the most recent outbreak from November 2020 to March 2021. In Hong Kong, the number of infections remains low for most of the time and the government has continuously imposed soft interventions. In the middle of November 2020, a newly imported case has triggered a new outbreak, resulting in a large $R_t$ value. However, the $R_t$ level returned to be around 1 very soon as the government has further tightened social distancing measures at that time. From late January 2021, Hong Kong started to implement mandatory lockdown in the restricted areas. Since then, the number of daily cases remains at low level. Compared with the results from EpiEstim and EpiNow2, the results from DARt have similar trend with the others. The delays in the results of EpiEstim are still significant. We also investigate the performance of DARt using different types of observations and present the results of $R_t$ estimation using onsets and confirmed cases in S1 Text.

Fig 4C shows the inference results from the United Kingdom's reported confirmed cases [35]. It is noted that the United Kingdom was one of the first countries in the world to authorise the emergency use of COVID-19 vaccines. Since early December 2020, the United Kingdom rolled out its COVID-19 mass vaccination programme. By mid-February 2021, the United Kingdom had successfully hit its target of 15 million first-dose COVID-19 vaccinations, encompassing the top four priority groups for vaccination. As of April 22, 2021, the UK had reached its target of 33 million (63% adults) first-dose COVID-19 vaccinations and 11 million (21% adults) second-dose. After 3 months since the mass vaccination programme started, the number of infection cases continuously followed a downward trend. However, since further easing of COVID-19 restrictions in mid-May 2021, $R_t$ gradually increased. During Euro 2020 football match (from June 11 to July 11), the $R_t$ value remained above 1. An immediate decrease in $R_t$ occurred when Euro 2020 was finished and since then $R_t$ remained around 1. The results from DARt, EpiEstim and EpiNow2 are generally consistent. However, DARt has accurately detected and responded to the impact of the completion of Euro 2020 in mid-July. In addition to studying the whole country, we applied DARt to typical cities in England to investigate the local epidemic dynamics as well (please refer to section 4 of S1 Text).

Fig 4D shows the inferred epidemic dynamics in Sweden from the daily reported data [36]. We find that the daily reported cases in Sweden had shown dramatic local fluctuations that were

likely to be caused by misreporting. The reported cases dropped to 0 on Saturdays, Sundays and Mondays. This kind of fluctuations in observations could induce unnecessary fluctuations to $R_t$ curves. Therefore, we used Sweden's data to further test the robustness of our scheme in the presence of undesirable local fluctuations in observations. The results suggested that the influence of such periodic fluctuations has been smoothed by DARt and EpiNow2 to yield a consistent $R_t$ curve, where results from EpiEstim have shown significant local fluctuations.

To summarise, DARt has been applied to four different regions for investigating the transmission dynamics of COVID-19 to demonstrate its real-world applicability and effectiveness. Consistent with the findings in the simulation study, DARt has shown its advantages in the following aspects: 1) Instantaneity—DARt adopts a window-free sequential Bayesian inference approach to detect and indicate abrupt epidemic changes; 2) Robustness—with Bayesian smoothing, the $R_t$ curve from DARt is stable at the presence of observation noise; 3) Temporal accuracy—DARt performs a joint estimation of $R_t$ and $j_t$ by explicitly encoding the lag into observation kernels.

## Discussion and conclusion

In this paper, we have proposed a Bayesian data assimilation scheme for estimating the time-varying epidemiological parameters based on observations. To study a real-world application scenario, we focus on estimating $R_t$ and provide a state-of-the-art $R_t$ estimation tool, DARt, supporting the study of a wide range of observations. In the DARt system, epidemic states can therefore be updated using newly observed data, following a data assimilation process in the framework of sequential Bayesian updating. For the model inference, a particle filtering/smoothing method is used to approximate the $R_t$ distribution in both forward and backward directions of time, ensuring the $R_t$ at each time step assimilates information from all time points. By taking the Bayesian approach, we have processed the uncertainty in $R_t$ estimation by accommodating observation uncertainty in likelihood mapping and introduced Bayesian smoothing to incorporate sufficient information from observations. Our method provides a smooth $R_t$ curve together with its posterior distribution. We have demonstrated that inferred $R_t$ curves can describe different observations accurately. Our work is not only important in revealing the epidemical dynamics but also useful in assessing the impact of interventions. The sequential inference mechanism of $R_t$ estimation takes into account the accuracy of time alignment and provides an abrupt change indicator. Different from approaches of directly incorporating interventions as co-factors into epidemic model [13,37], our method offers a promising method for intervention assessment.

We have made some approximations to facilitate the implementation. First, the observation time and generation time distributions are truncated into fixed and identical lengths. Theoretically, these two distributions can be of any length, while most values are quite small in practice. In our state transition model, one variable of the latent state is a vectorised form of infection numbers over a period. The purpose of vectorisation is to facilitate implementation by making the transition process Markovian. The length of this vector variable is determined by the length of effective observation time and the generation time distributions. Truncating these two distributions to a limited length, by discarding small values, would facilitate vectorisation. Apart from truncation, we have assumed that these two distributions do not change during the prevalence of disease. However, as we have discussed previously [23], introducing interventions, such as an increased testing capacity, would affect the observation time. The distribution of generation time would also change as the virus is evolving. It is possible to extend our model by adding a time-varying observation function. For example, the testing capability and time-varying mortality rate could also be considered in the observation process.

Second, we approximate the variance of observation error empirically. Given that variance of observations is unknown and could change over time across different regions, the standard deviation of the Gaussian likelihood function is not set to a fixed value in our scheme. Instead, we estimate the region-specific time-varying observation variance from the observational data. Although the empirical estimation yielded reasonable results for the four regions and cities in the UK (see S1 Text), it may generate some implausible results in some scenarios, for example, when the epidemic is growing or resurging explosively, leading to an overestimation of observation variance. An adaptive error variance inference should be made to tackle this issue.

The third approximation is implicit in the use of a particle filter to approximate the posterior distributions over model state variables–including $R_t$–with a limited number of samples (i.e., particles). Particle filtering makes no assumption about the form of posterior distributions. On the contrary, the variational equivalent of the particle filter, namely variational filtering [38] provides an analytical approximation to the posterior probability and can be regarded as limiting solutions to an idealised particle filter, with an infinite number of particles [39]. Considering the importance of both the mean value of $R_t$ and its estimation uncertainty for advising governments on policymaking, an analytical approximation is desirable to help properly quantify uncertainty.

Finally, change detection is approximated by the change indicator $M_t$, which is included as part of the latent state and inferred during particle filtering. This work opens an avenue to explore variational Bayesian inference for switching state models [40]. Crucially, variational procedures enable us to assess model evidence (a.k.a. marginal likelihood) and hence allow automatic model selection. Examples of Variational Bayes and model comparison to optimise the parameters and structure of epidemic models can be found in previous studies [41]. These variational procedures can be effectively applied to change detection.

In conclusion, our work provides a practical scheme for accurate and robust estimation of time-varying epidemiological parameters. It opens a new avenue to study epidemic dynamics within the Bayesian data assimilation framework. We provide an open-source $R_t$ estimation package as well as an associated Web service that may facilitate other people's research in computational epidemiology and the practical use for policy development and impact assessment.

## Methods

The proposed Bayesian data assimilation framework for estimating epidemiological parameters include three main components: 1) a **state transition model**—describing the evolution of the latent state; 2) an **observation function**–defining an observation process and describing the relationship between the latent state and observations; 3) a **sequential Bayesian engine**: estimating statistical reason time-varying model parameters with uncertainty by assimilating prior state information provided by the transition model and the newly available observation. In this section, we introduce a real-world application of the proposed data assimilation framework to estimate one of the key epidemiological parameters, $R_t$. The modelling epidemic dynamics is characterised by the renewal process, which is the foundation of our state transition model. We then describe the observation function, linking a sequence of infection numbers with the observation data. Next, we present a detailed state transition model and propose the sequential Bayesian update module.

### 1. Renewal process for modelling epidemic dynamics

Common $R_t$ estimation methods include compartment model-based methods (e.g., SIR and SEIR [42]) and time-since-infection models based on renewal process [31]. Their relationships

are discussed in Section 1.1 of S1 Text. Comparative studies have been conducted in [21] to show that EpiEstim, one of the renewal process-based methods, outperforms other methods in terms of accuracy and timeliness. Given the renewal process, the key transition equation derived from the process is:

$$j_t = R_t \sum_{k=1}^{T_w} w_k j_{t-k} \tag{1}$$

where $j_t$ is the number of incident infection cases on day $t$, $T_w$ is the time span of the set $\{w_k\}$, and individual $w_k$ is the probability that the secondary infection case occurs $k$ days after the primary infection, describing the distribution of generation time [10]. The profile of $w_k$ is related to the biological characteristics of the virus and is generally assumed to be time-independent during the epidemic. Considering the simplicity and superior performance of applying the renewal process to model epidemic dynamics, our work adopts Eq (1) as the basic transition function for joint estimation of $R_t$ and $j_t$.

## 2. Observation process

In epidemiology, the daily infection number $j_t$ cannot be measured directly but is reflected in observations such as the case reports of onset, confirmed infections and deaths. There is an inevitable time delay between the real date of infection and the date reporting, due to the incubation time, report delay, etc. Taking account of this time delay, we model the observation process as a convolution function between kernel $\varphi$, and the infection number in $T_H$ most recent days.

$$C_t = \sum_{k=d}^{T_H} \varphi_k j_{t-k} \tag{2}$$

where $C_t$ is the observation data, and $\varphi_k$ is the probability that an individual infected is detected on day $k$. $T_H$ is the maximum dependency window. It is assumed that the past daily infections before this window do not affect the current observation $C_t$. Since there is a delay between observation and infection, we suppose the most recent infection that can be observed by $C_t$ is at the time $t-d$, where $d$ is a constant determined by the distribution of observation delay.

To accommodate various observation types (e.g., the number of daily reported cases, onsets, deaths and infected cases), DARt will choose the appropriate time delay distributions accordingly. For example, for the input of onsets, the infected-to-onsets time distribution is chosen to be the kernel in the observation function. For the input of daily reported cases, the infected-to-onset and the onset-to-report delays are used together as the kernel in the observation function. These delay distributions can be either directly obtained from literature or inferred from case reports that contain individual observation delays [3]. Detailed descriptions of the observation functions for different epidemic curves can be found in S1 Text (Section 1.2).

## 3. Sequential Bayesian Inference

In Fig 5, we illustrate the Bayesian inference scheme of DARt with the following details.

**State transition model.** In our model, indirectly observable variables $j_t$ and $R_t$ are included in the latent state. The state transition function for $R_t$ is commonly assumed to follow a Gaussian random walk [18] or constant within a sliding window as implemented in EpiEstim. Such a simplification cannot capture an abrupt change in $R_t$ under stringent intervention measures. To capture such abrupt changes, we introduce an auxiliary binary latent variable $M_t$ to indicate the switching dynamics of the epidemiological parameters under interventions

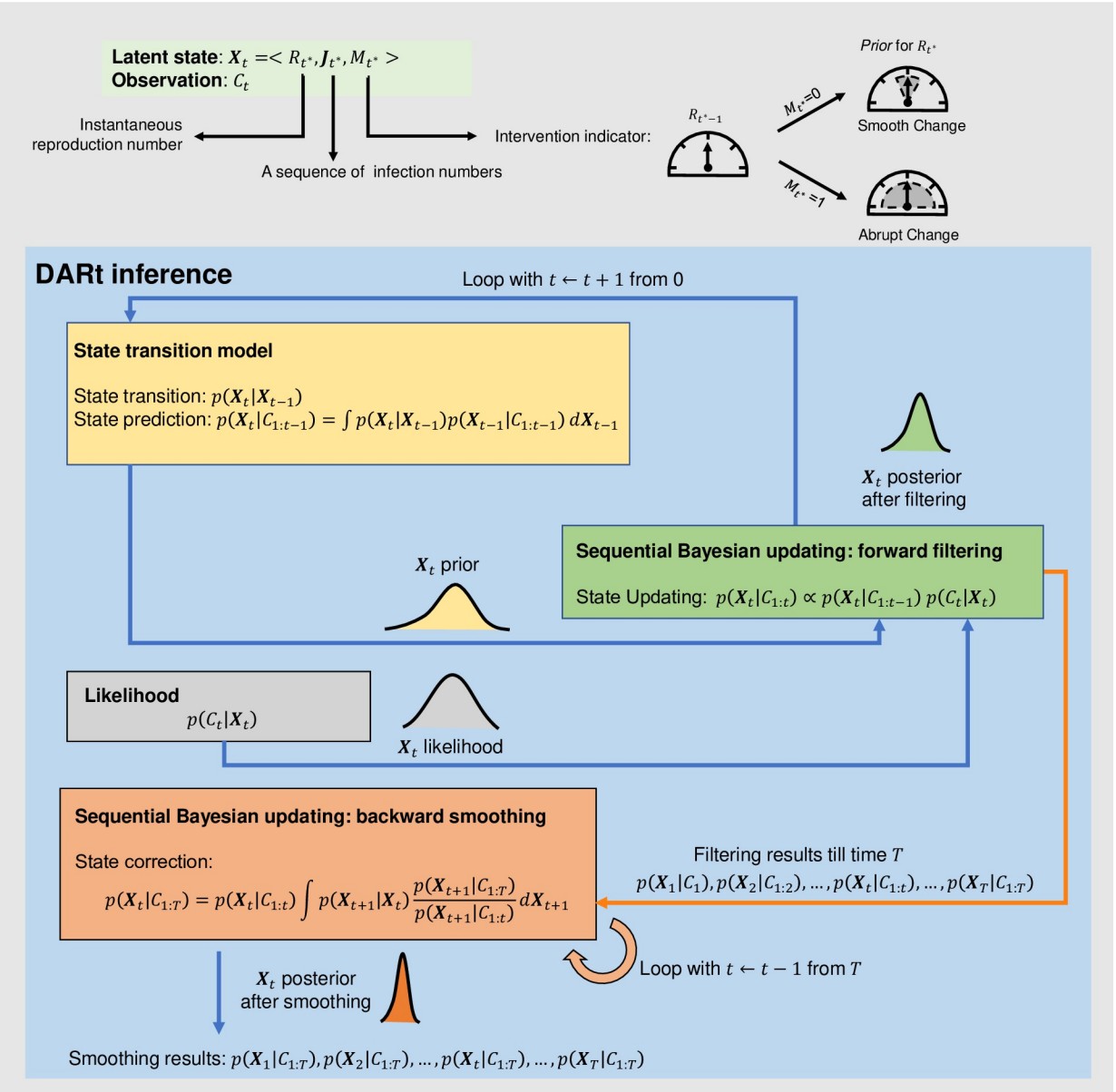

**Fig 5. Three components of DARt inference model: state transition model, observation function and sequential Bayesian update module with two phases (forward filtering and backward smoothing).** The latent state that can be observed in $C_t$ are defined as $X_t = <R_{t^*}, J_{t^*}, M_{t^*}>$ where $R_{t^*}$ is the instantaneous reproduction number, $M_{t^*}$ is a binary state variable indicating different evolution patterns of $R_{t^*}$, $J_{t^*} = [j_{t^*-T_\varphi+1}, j_{t^*-T_\varphi+2}, \ldots, j_{t^*}]$ is a vectorised form of infection numbers $j_t$, $t^*$ indicates the most recent infection that can be detected at time $t$ is from the time $t^*$ due to observation delay, and $T_\varphi$ is the length of the vector $J_{t^*}$ such that $C_t$ is only relevant to $J_{t^*}$ and $j_{t^*+1}$ only depends on $J_{t^*}$ via the renewal process.

without assuming a pre-defined evolution pattern (e.g., constant or exponential decay). $M_t = 0$ indicates a smooth evolution corresponding to minimal or consistent interventions; $M_t = 1$ indicates an abrupt change of corresponding to new interventions or outbreak. The smooth evolution is modelled as a Gaussian random walk while the abrupt change is captured through resetting the parameter memory by assuming a uniform probability distribution for the next time step of estimation. Doing so provides an automatic way of framing a new epidemic period

that was manually done in [13]. The transition of $M_t$ is modelled as a discrete Markovian process with fixed transition probabilities controlling the sensitivity of change detection:

$$p(R_t|R_{t-1}, M_t) \sim \begin{cases} \mathcal{N}(R_{t-1}, \sigma_R^2) \ M_t = 0 \ \text{Mode I} \\ \mathrm{U}[0, R_{t-1} + \Lambda] \ M_t = 1 \ \text{Mode II} \end{cases} \tag{3}$$

where $\mathcal{N}(R_{t-1}, \sigma_R^2)$ is a Gaussian distribution with the mean value of $R_{t-1}$ and variance of $\sigma_R^2$, describing the random walk with the randomness controlled by $\sigma_R$. $\mathrm{U}[0, R_{t-1}+\Lambda]$ is a uniform distribution between 0 and $R_{t-1}+\Lambda$ allowing sharp decrease while limiting the amount of increase. This is because we assume that $R_t$ can have a significant decrease when intervention is introduced but it is unlikely to increase dramatically as the characteristics of the disease would not change instantly.

The transition of the change indicator $M_t$, is modelled as a discrete Markovian process with fixed transition probabilities:

$$p(M_t = 0|M_{t-1}) = p(M_t = 0) = \alpha \tag{4A}$$

$$p(M_t = 1|M_{t-1}) = p(M_t = 1) = 1 - \alpha \tag{4B}$$

where $\alpha$ is a value close to and lower than 1. The above function means that the value of $M_t$ is independent of $M_{t-1}$, while the probability of Mode II (i.e., $M_t = 1$) is quite small. This is because it is unlikely to have frequent abrupt changes in $R_t$.

For the incident infection $j_t$, the state transition can be modelled based on Eq (1) as $p(j_t|j_{t-1}, \ldots, j_{t-T_w})$. To make the transition process Markovian, we vectorise the infection numbers as follows. Suppose the infection numbers $\{j_{t-k}\}_{k=d}^{T_H}$ that can be observed in $C_t$ are all included in $\boldsymbol{J}_{t^*} = [j_{t^*-T_\varphi+1}, j_{t^*-T_\varphi+2}, \ldots, j_{t^*}]$, where $t^* = t-d$, and the length of this vector $T_\varphi$ is larger than or equal to $T_H-d+1$. We also require $T_\varphi$ to be not smaller than $T_w$. Therefore, all the historical information needed to infer $j_{t^*}$ is available from $\boldsymbol{J}_{t^*-1}$, i.e., $\boldsymbol{J}_{t^*}$ only depends on $\boldsymbol{J}_{t^*-1}$ (i.e., being Markovian). The state transition process and observation process are illustrated in Fig 6.

The latent state in our model is then defined as $\boldsymbol{X}_t = <R_{t^*}, \boldsymbol{J}_{t^*}, M_{t^*}>$, which contribute to $C_t$ at time $t$. The state transition function of $\boldsymbol{J}_{t^*}$ is therefore Markovian:

$$p(\boldsymbol{J}_{t^*}|\boldsymbol{J}_{t^*-1}, R_{t^*}) = Poisson(j_{t^*}; \ R_{t^*} \sum_{k=1}^{T_w} w_k j_{t^*-k}) \prod_{m=1}^{T_\varphi-1} \delta(\boldsymbol{J}_{t^*}^{(m)}, \boldsymbol{J}_{t^*-1}^{(m+1)}) \tag{5}$$

where $\boldsymbol{J}_{t^*}^{(m)}$ is the $m$-th component of the latent variable $\boldsymbol{J}_{t^*}$ and $\delta(x, y)$ is the Kronecker delta function (please refer to S1 Text for more details). With Eqs (3)–(5), the latent state transition function $p(\boldsymbol{X}_t|\boldsymbol{X}_{t-1})$ can be obtained as a Markov process:

$$p(\boldsymbol{X}_t|\boldsymbol{X}_{t-1}) = p(\boldsymbol{J}_{t^*}|\boldsymbol{J}_{t^*-1}, R_{t^*})p(R_{t^*}|R_{t^*-1}, M_{t^*})p(M_{t^*}|M_{t^*-1}) \tag{6}$$

**Forward filtering.** We formulate the inference of the latent state $\boldsymbol{X}_t = <R_{t^*}, \boldsymbol{J}_{t^*}, M_{t^*}>$ with the observations $C_t$ as within a data assimilation framework. A sequential Bayesian filtering approach is adopted to infer the time-varying latent state, which updates the posterior estimation using the latest observations following Bayes' rule. This approach differs from the fixed prior in the Bayesian inference of static parameters. This filtering mechanism computes the posterior distribution of the latent state by assimilating the forecast from the forward transition model with the information from the new epidemiological observations. For the

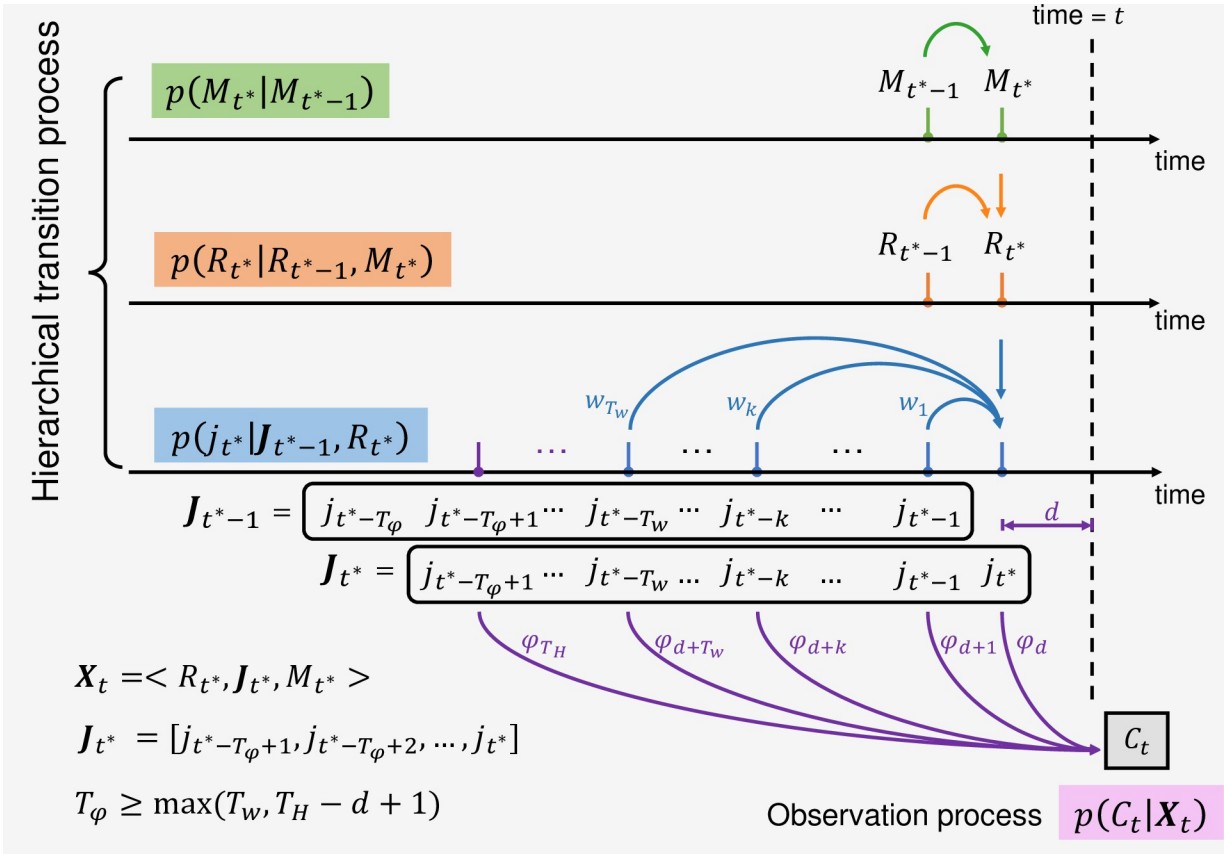

**Fig 6. Illustration of the hierarchical transition process and observation process.** The most recent infection that can be observed by $C_t$ is at the time $t^* = t-d$ where $d$ is a constant determined by the distribution of observation delay. Suppose $T_\varphi$ is the length of the vector $\boldsymbol{J}_{t^*} = [j_{t^*-T_\varphi+1}, j_{t^*-T_\varphi+2}, \ldots, j_{t^*}]$ such that $C_t$ is only relevant to $\boldsymbol{J}_{t^*}$ and $j_{t^*}$ only depends on $\boldsymbol{J}_{t^*-1}$ via the renewal process. Therefore, $T_\varphi \geq \max(T_w, T_H-d+1)$. The case that $T_\varphi = T_H-d+1$ is depicted in this figure.

implementation of this Bayesian updating process, we adopt a particle filter method [26] to efficiently approximate the posterior distribution through Sequential Monto Carlo (SMC) sampling. This eschews any fixed-form assumptions for the posterior–of the sort used in variational filtering and dynamic causal modelling [38].

Let us denote the observation history between time 1 and $t$ as $C_{1:t} = [C_1, C_2, \ldots, C_t]$. Given previous estimation $p(\boldsymbol{X}_{t-1}|C_{1:t-1})$ and new observation $C_t$, we would like to update the estimation of $\boldsymbol{X}_t$, i.e., $p(\boldsymbol{X}_t|C_{1:t})$ following Bayes' rule with the assumption that $C_{1:t}$ is conditionally independent of $C_{1:t-1}$ given $\boldsymbol{X}_t$:

$$p(\boldsymbol{X}_t|C_{1:t}) = \frac{p(C_t|\boldsymbol{X}_t)p(\boldsymbol{X}_t|C_{1:t-1})}{\int p(C_t|\boldsymbol{X}_t)p(\boldsymbol{X}_t|C_{1:t-1})d\boldsymbol{X}_t} \tag{7}$$

where $p(\boldsymbol{X}_t|C_{1:t-1})$ is the prior and $p(C_t|\boldsymbol{X}_t)$ is the likelihood. The prior can be written in the marginalised format:

$$p(\boldsymbol{X}_t|C_{1:t-1}) = \int p(\boldsymbol{X}_t|\boldsymbol{X}_{t-1})p(\boldsymbol{X}_{t-1}|C_{1:t-1})d\boldsymbol{X}_{t-1} \tag{8}$$

where $X_t$ is assumed to be conditionally independent of $C_{1:t-1}$ given $X_{t-1}$, and the transition $p(X_t|X_{t-1})$ is defined in Eq (6) based on the underlying renewal process. The likelihood $p(C_t|X_t)$ can be calculated assuming the observation uncertainty follows a Gaussian distribution:

$$p(C_t|X_t) \sim \mathcal{N}(H(X_t), \sigma_C^2) \tag{9}$$

where $H$ is the observation function with a kernel chosen according to the types of observations and $\sigma_C^2$ is the variance of observation error estimated empirically. To show the benefits of using this Gaussian likelihood function, we show the simulation results of using Poisson likelihood without considering the observation noise. Results can be found in Fig B in S1 Text, where the estimations fluctuate dramatically under noisy observation.

By substituting Eq (8) into Eq (7), we obtain the iterative update of $p(X_t|C_{1:t})$ given the transition $p(X_t|X_{t-1})$ and likelihood $p(C_t|X_t)$:

$$p(X_t|C_{1:t}) = \frac{p(C_t|X_t) \int p(X_t|X_{t-1})p(X_{t-1}|C_{1:t-1})dX_{t-1}}{\iint p(C_t|X_t) \int p(X_t|X_{t-1})p(X_{t-1}|C_{1:t-1})dX_{t-1}dX_t} \tag{10}$$

**Backward smoothing.**   The estimated result $p(X_t|C_{1:t})$ from aforementioned forward filtering only includes the past and present information flows, corresponding to the prior $p(X_t|C_{1:t-1})$ and likelihood $p(C_t|X_t)$, respectively. The filtering estimates would be accurate if all related infections are fully observed in $C_{1:t}$. However, this is certainly not the case due to observation delay. In order to reduce the uncertainty from forward filtering, we adopt the Bayesian backward smoothing technique, estimating the latent state at a time $t$ retrospectively, given all observations available till time $T$ ($T>t$). Compared with other parameter estimation methods [13], Bayesian data assimilation takes advantage of additional information to smooth inference results with reduced uncertainty caused by incomplete observations. More specifically, the smoothing mechanism can be described as: given a sequence of observations $C_{1:T}$ up to time $T$ and filtering results $p(X_t|C_{1:t})$, for all time $t<T$, the state estimates are smoothed as:

$$p(X_t|C_{1:T}) = p(X_t|C_{1:t}) \int p(X_{t+1}|X_t) \frac{p(X_{t+1}|C_{1:T})}{p(X_{t+1}C_{1:t})} dX_{t+1} \tag{11}$$

where $p(X_{t+1}|C_{1:T})$ is the smoothing results at time $t+1$ where $\int p(X_{t+1}|X_t) \frac{p(X_{t+1}|C_{1:T})}{p(X_{t+1}|C_{1:t})} dX_{t+1}$ is the smoothing factor. In this way, all the relevant observations are fully exploited to enable us to reduce the uncertainty of parameter estimation. Comparing with the sliding-window (i.e., averaging inference) approaches, our sequential Bayesian updating with backward smoothing mechanism features an instantaneous epidemiological parameter estimation and smoothing uncertainty through the utilisation of all available observations. More details can be found in S1 Text.

## Data

We obtained daily onset or confirmed cases of four different regions (Wuhan, Hong Kong, Sweden, UK) from publicly available sources [1,34–36]. For Wuhan, we adopted the daily number of onset patients from the retrospective study [1] (from the end of December 2020 to early March 2020). For UK data, we downloaded the daily report cases (cases by date reported) from the official UK Government website for data and insights on Coronavirus (COVID-19) [35] (from the start of January 2021 to the end of August 2021) accessed on 30th of August 2021. Data for UK Cities were also downloaded from the same source [35] (from the start of January 2021 to the start of September 2021) accessed on 2nd of September 2021. For Sweden

data, we downloaded the daily number of confirmed cases from the Our World in Data COVID-19 dataset [36] (from the middle of January 2021 to the start of September 2021) accessed on $2^{nd}$ of September 2021. For Hong Kong, we downloaded the case reports from government website [34] (from the end of November 2020 to the end of March 2021), including descriptive details of individual confirmed case of COVID-19 infection in Hong Kong. For those asymptomatic patients whose onset date are unknown, we set their onset date as their reported date, and for those whose onset date is unclear, we simply removed and neglected these records. Only local cases and their related cases are considered, while imported cases and their related cases are excluded.

We release DARt as open-source software for epidemic research and intervention policy design and monitoring. The source codes of our method and our web service are publicly available online (https://github.com/Kerr93/DARt).

## Supporting information

**S1 Text. Supplementary document containing some supporting information.** Fig A. Illustrations of three types of observations and corresponding distributions of delay from the real infection date and observation. Fig B. Comparison between the simulation results using Poisson likelihood and Gaussian likelihood in DARt (both with 95% CrI). Fig C. Comparison between the estimated daily infection. The estimated infection by DARt is drawn in black with 95% CrI. The ground-truth simulated infection is in red and the back calculated infection is in yellow. Fig D. Comparison of estimated $R_t$ curves of Hong Kong using different observations. Subplot A) shows $R_t$ estimations (in black) from confirmed cases (in yellow). Subplot B) shows $R_t$ estimations (in black) from daily onset (in yellow). Fig E. Epidemic dynamics in London, Leicester, Birmingham, Liverpool, Manchester, Sheffield, and Leeds. The top row of each subplot shows the number of daily observations (in yellow), the estimated daily observations (in blue) and the estimated daily infections (in green). The middle row shows the DARt results of $R_t$ curve with 95% CrI (in black), while the probability of having abrupt changes is shown in the bottom row (i.e., $M_t = 1$) (in green). Fig F. The $R_t$ estimation results under different levels of observation noise: A) N = 0, B) N = 1, C) N = 2 and D) N = 3, where the added Gaussian noise has the standard deviation equal to $N$ times of the unperturbed observation. Fig G. The $R_t$ estimation results of DARt with different truncation threshold: A) 0.01, B) 0.05 and C) 0.1. Fig H. A) The $R_t$ estimation results of DARt obtained from the generation time and observation delay distributions with uncertainties. B) The $R_t$ estimation results of DARt obtained from the generation time distribution following a Lognormal distribution. Table A. Transition probabilities of $M_{t^*}$. Table B. Simulation results using synthetic data in the main manuscript. $\Delta R_t$-mean/$\Delta J_t$-mean and $\Delta R_t$-sd/$\Delta J_t$-sd are the mean and standard deviation of the differences between synthetic $R_t/J_t$ and estimated $R_t/J_t$. Since EpiEstim does not estimate $J_t$, we leave the corresponding values as NA.
(PDF)

## Acknowledgments

Dr Simon Wang, at the Language Centre, HKBU, has helped improve the linguistic presentation of this manuscript.

## Author Contributions

**Conceptualization:** Shuo Wang, Ling Li, Richard Yi Da Xu, Karl J. Friston, Yike Guo.

**Data curation:** Yuting Xing.

**Formal analysis:** Xian Yang, Shuo Wang.

**Methodology:** Xian Yang, Shuo Wang, Richard Yi Da Xu, Karl J. Friston, Yike Guo.

**Software:** Xian Yang, Yuting Xing.

**Supervision:** Yike Guo.

**Writing – original draft:** Xian Yang, Shuo Wang, Yike Guo.

**Writing – review & editing:** Xian Yang, Shuo Wang, Yuting Xing, Ling Li, Richard Yi Da Xu, Karl J. Friston, Yike Guo.

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
