## [Decision Letter · Decision Letter 0]

6 Aug 2021

Dear Dr Yang,

Thank you very much for submitting your manuscript "Bayesian data assimilation for estimating transmission dynamics in computational epidemiology" for consideration at PLOS Computational Biology.

As with all papers reviewed by the journal, your manuscript was reviewed by members of the editorial board and by several independent reviewers. In light of the reviews (below this email), we would like to invite the resubmission of a significantly-revised version that takes into account the reviewers' comments.

We cannot make any decision about publication until we have seen the revised manuscript and your response to the reviewers' comments. Your revised manuscript is also likely to be sent to reviewers for further evaluation.

Sincerely,

Alison L. Hill

Associate Editor

PLOS Computational Biology

Rob De Boer

Deputy Editor

PLOS Computational Biology

Reviewer's Responses to Questions

**Comments to the Authors:**

Reviewer #1: The review is uploaded as an attachment

Reviewer #2: A summary of my comments (detailed below):

The method described in this paper is interesting and promising, given its success at handling the issues it sets out to address (lags, uncertainty and averaging), and the demonstrated comparisons to results output from EpiEstim

It is necessary to add a thorough comparison to EpiNow

The deconvolution discussion and comparisons should be extended, and the accuracy of Supplementary Figure 3 should be carefully checked (and commented on, depending on the results of the check)

This paper needs to consider uncertainty from potential misspecification of the generation interval distribution and the observation delay distribution

The analysis of the sensitivity of the method to truncation needs to be separate from the issue of adding variance to the estimates of the parameters for the generation interval distribution and observation delay distribution

Some issues with writing, organization, and figures should be fixed

The instantaneous reproductive number Rt has become a ubiquitous tool for assessing the state of the COVID-19 pandemic in different regions across the globe. It has been used to inform policy decisions such as whether to impose or lift restrictions like lockdowns and mask mandates, and to determine the effect of such interventions.

Yet computing Rt from the data that is typically available to researchers during the course of the pandemic is nontrivial because of the forward-looking nature of the interpretation of Rt as the average number of secondary infections a person who is infectious at time t will produce in the next timestep, a measure which approximates the number of secondary infections a person infected at time t will produce over the course of their infection using only data accessible up to time t. Moreover, Rt computations depend on epidemiological parameters that are often unknown or even unknowable in real analyses.

Although best practices for computing Rt have been proposed since the beginning of the pandemic (Gostic et all 2020, e.g.), several problems have remained as important challenges to computing Rt accurately. These include the lag between time of infection and measured case numbers from testing, hospitalization, or deaths; the difficulty in properly accounting for multiple sources of uncertainty; and the ambiguity in selecting a window size for averaging in computing Rt using what are currently state of the art methods.

This paper advances the current state of the field by proposing a novel Bayesian method, DARt, for addressing these three problems using the technique of data assimilation. It uses simulated data to demonstrate advantages of this method compared to methods for computing Rt using the package EpiEstim, which is currently a widely used, recommended method for computing Rt in real applications. One particularly interesting feature of the method is how it handles abrupt changes in Rt through the incorporation of a binary latent variable that infers the probability of large changes in Rt in short periods of time.

This paper is not the first, however, to implement a Bayesian method with the goal of addressing lagged data, uncertainty and averaging in computations of Rt. In particular, the package EpiNow (and its more recent iteration EpiNow2), developed by Abbott et al 2020, implements another Bayesian method meant to to solve these issues within the context of best practices for computing Rt. While the authors of this paper briefly mention EpiNow2, they do not compare their methods for inferring the curve of incident infections from the curve of observations and Rt to the methods used by EpiNow2 in detail. Given the importance of EpiNow2 as a highly developed tool for solving problems this paper is addressing as well, the authors’ choice to compare only to EpiEstim and not to EpiNow2 is a major oversight that prevents the researchers from putting their work in context with currently available tools.

The paper also deemphasizes the importance of the Goldstein et al deconvolution method for taking into account lags in Rt calculations, comparing DARt to deconvolution only in section 2.3 of the supplementary information. Deconvolution has the serious disadvantages of not allowing for computation of uncertainty in the inference of the incident infection curve and requiring a window size choice in its calculation of Rt, but it is typically very successful at producing a point estimate of the infection curve from the curve of observations when the distributions of generation interval and delays to observation are assumed to be known and with an appropriate choice of stopping criterion for the Richardson-Lucy algorithm.

The lack of agreement between the simulated infection curve and the deconvolved infection curve in Supplementary Figure 3 is surprising and may indicate an issue with the implementation of the deconvolution code. Such an issue can occur if the indexing of time in the delay distribution is off compared to the indexing of time in the vectorized curve of observations (one must be careful to keep track of whether the delay distribution before truncation mathematically begins with the probability that the delay equals 0 days vs. the delay equaling 1 day; such an off-by-one error can cause disagreements like the one shown here - in particular, the disagreement between the location of the peak of the deconvolved curve and the true infection curve - although it is not the only possible issue). This disagreement should be carefully checked; if the deconvolution is truly not reproducing the peak correctly, this requires at least a comment on the intuition for why this is happening.

This paper also neglects a source of uncertainty that is very important to real-life calculations of Rt. Namely, the observation delay distribution is very rarely known exactly (or even very accurately at all); even the generation interval distribution can be unknown, for example in the case of new COVID-19 variants. The authors provide a sensitivity analysis in supplementary figure 5 that is meant to address this issue, but the analysis is insufficient for a few reasons. First, it seems to test the effects of different thresholds for the truncation and samples from generation time distribution and observation delay distribution with variance in the parameters in a single plot. Second, there is no demonstrated analysis of the sensitivity of the method to cases where the shape of the assumed generation interval and/or observation delay are misspecified (with the wrong shape/scale/logmean/variance compared to the generation interval and observation delay used to generate the observation curve). Such misspecification can, at least for some methods, severely impact the accuracy of the inferred infection and Rt curves. Ideally, a method for inferring the infection curve and Rt with uncertainty would take into account uncertainty derived from the likely misspecification of generation interval and observation delay distribution in real applications; barring this, which is difficult to model, the sensitivity analysis should at least be modified to: a) test the effects of delay distribution minimum probability threshold and uncertainty in the delay distributions separately, and b) test the sensitivity of the method to cases where the delay distributions are misspecified, not just uncertain.

There are a few more minor issues that need to be addressed. In no particular order:

There are some scattered language and grammatical errors which need to be edited.

The arbitrary choice of Delta in the prior for Mt is not ideal, especially since the decision to allow Rt to increase discontinuously for the Hong Kong analysis (and the constraint that Rt not increase discontinuously in other cases) is inserted “by hand” rather than “discovered” by means of the inference

The extended description of the algorithm in the “Results” section seems potentially misplaced; I would suggest considering making this discussion much more concise and putting more of these details in the Methods section

In figure 3, c) and d) should have the same scale on the y axis

In the Sweden analysis of Figure 6, in my opinion it is somewhat disingenuous to directly input the raw data into EpiEstim with small window sizes, as in reality, a reasonable researcher would always try to smooth out the zeros and obvious reporting errors in this dataset before entering into EpiEstim (of course, the difference is that this smoothing would be by hand and thus require more of an arbitrary choice, whereas the described method does this without arbitrary choices)

**Have the authors made all data and (if applicable) computational code underlying the findings in their manuscript fully available?**

Reviewer #1: Yes

Reviewer #2: Yes

PLOS authors have the option to publish the peer review history of their article (what does this mean?). If published, this will include your full peer review and any attached files.

Reviewer #1: No

Reviewer #2: No
---

## [Decision Letter · Decision Letter 1]

18 Nov 2021

Dear Dr Yang,

Thank you very much for submitting your manuscript "Bayesian data assimilation for estimating epidemic evolution: a COVID-19 study" for consideration at PLOS Computational Biology. As with all papers reviewed by the journal, your manuscript was reviewed by members of the editorial board and by several independent reviewers. The reviewers appreciated the attention to an important topic. Based on the reviews, we are likely to accept this manuscript for publication, providing that you modify the manuscript according to the review recommendations.

Sincerely,

Alison L. Hill

Associate Editor

PLOS Computational Biology

Rob De Boer

Deputy Editor

PLOS Computational Biology

[LINK]

Reviewer's Responses to Questions

**Comments to the Authors:**

Reviewer #1: The review is uploaded as an attachment

Reviewer #2: The main contributions and importance of the paper have not changed since the original submission; the comments below are specific to this submission.

First, I think the sensitivity analyses in section 6 of the Supplement are a nice addition in response to previous comments from the reviewers.

Major comment:

1. The comparison between DARt, EpiNow, and EpiEstim is no longer "apples to apples," as both DARt and EpiNow are implemented taking into account an observation delay, but EpiEstim is now implemented as "plug and play" with no consideration of observation delay. In the first submission, EpiEstim was used in conjunction with Richardson-Lucy deconvolution to take into account observation delays in computing the incident infections before computing Rt. If the authors want to separate the issue of inferring incident infections using deconvolution from the accuracy of the computation of Rt itself in their application of EpiEstim, they could still take into account observation delays to align time values properly using a more simplistic method such as subtracting the mean or median of the observation delay distribution from the reported times. Some sort of correction for the observation delay still needs to be made for the EpiEstim Rt calculations as otherwise issues with inferring the shape of Rt are intermixed with issues with inferring the timing in Rt in EpiEstim but not in the other two comparisons, making conclusions about the comparison with EpiEstim misleading.

Minor comments:

2. The main text's writing needs to be significantly edited. Although the authors definitely made an effort to address writing issues since the first submission, this submission still has many grammatical and language issues throughout. The text of the Supplement is written much more clearly.

3. The word "easement" needs to be replaced in the main text and supplement (say, with "easing" of restrictions) because its meaning is currently misused.

4. Overall, moving the details of the algorithm from the "Results" to the "Methods" as suggested in the first round of reviews has improved the flow and readability; however, now some aspects of the "Discussion" don't make sense, as they aren't explained in the text until after the Discussion section. Specifically:

- particle filtering needs to be mentioned in the algorithm summary in "Results" in order for it to be mentioned in the "Discussion and Conclusion" section

- same for the definition of M_t

5. Replace "the Bayes rule" with "Bayes' rule" throughout

6. I still feel that the discussion of reporting issues with the Sweden data has issues (lines 364-371). The periodicity and predictability of the under-reporting and subsequent correction are not the same as random noise, so I don't think it is appropriate to say that the Sweden comparison shows robustness to "observation noise".

7. The formatting of citations in the text is awkward. For example, when multiple citations are listed[A], [B-E], the comma outside the bracket is unclear. One way of fixing this would be to instead write[A, B-E], all in brackets. Moreover, although in the text usually there is no space between the text and the citation[example], sometimes there is a space [example 2]. Please check this for consistency throughout.

8. Throughout the text and supplement the word "synthesised" is used where the word "synthetic" would be more appropriate/conventional.

9. It's interesting that the EpiNow2 results are consistently over-smoothed. Is this because the analysis uses default parameters or is it impossible to improve this even by choosing parameters more deliberately? Please at least comment on this in a sentence when discussing EpiNow results (preferably, include analysis in a supplemental figure).

10. Dates need to be consistently formatted - for example in line 317, "21st of Jan. 2020" is different from later date formatting. Saying "Jan. 21, 2020" or "1/21/2020" would be more in line with common usage.

11. Consider providing the Supplement as a .pdf: Since the Supplement is provided in a .docx instead of .pdf as it stands, much of the formatting is lost when I open it on my machine.

12. In Supplementary Table 2, Rt-mean, Rt-sd, Jt-mean and Jt-sd are not the best notation because these represent *differences* between the synthetic and estimated values. I would suggest using notation that indicates this, such as $\\Delta$Rt-mean, $\\Delta$Rt-sd, etc.

13. In Supplementary Table 2, I expect that the EpiEstim row will change if an effort is made to take into account observation delays, as suggested in the major comment.

Additional Editorial comments:

We suggest the authors develop a title that better reflects the focus of the study. Otherwise, interested readers may not be able to find this paper. We advise against using the word "evolution" in the title, since in biology that generally means the Darwinian process of mutation and selection, which is not modeled here. Since you have specifically focused on a method for estimating Rt during emerging epidemics, we would recommend replacing this word with "growth" or "real-time reproduction number" or something similar. For example, "Bayesian data assimilation for estimating effective reproduction numbers during epidemics: applications to COVID-19"Similar comments apply to the wording used in the abstract.In the abstract, symbols should not be used before they are defined. Please use words instead of symbol for "Rt" when it's first introduced**********

**Have the authors made all data and (if applicable) computational code underlying the findings in their manuscript fully available?**

Reviewer #1: Yes

Reviewer #2: Yes

PLOS authors have the option to publish the peer review history of their article (what does this mean?). If published, this will include your full peer review and any attached files.

Reviewer #1: No

Reviewer #2: No

Figure Files:

Data Requirements:

Reproducibility:

References:

---

## [Editor Report · Decision Letter 2]

5 Jan 2022

Dear Dr Yang,

We are pleased to inform you that your manuscript 'Bayesian data assimilation for estimating instantaneous reproduction numbers during epidemics: applications to COVID-19' has been provisionally accepted for publication in PLOS Computational Biology.

Best regards,

Alison L. Hill

Associate Editor

PLOS Computational Biology

Rob De Boer

Deputy Editor

PLOS Computational Biology

---

## [Editor Report · Acceptance letter]

16 Feb 2022

PCOMPBIOL-D-21-00782R2 

Bayesian data assimilation for estimating instantaneous reproduction numbers during epidemics: applications to COVID-19

Dear Dr Guo,

I am pleased to inform you that your manuscript has been formally accepted for publication in PLOS Computational Biology. Your manuscript is now with our production department and you will be notified of the publication date in due course.

With kind regards,

Olena Szabo
